# Numerical and Experimental Analyses of Three-Dimensional Unsteady Flow around a Micro-Pillar Subjected to Rotational Vibration

**DOI:** 10.3390/mi9120668

**Published:** 2018-12-17

**Authors:** Kanji Kaneko, Takayuki Osawa, Yukinori Kametani, Takeshi Hayakawa, Yosuke Hasegawa, Hiroaki Suzuki

**Affiliations:** 1Faculty of Science and Engineering, Chuo University, Tokyo 112-8551, Japan; kaneko@nano.mech.chuo-u.ac.jp (K.K.); hayaka-t@mech.chuo-u.ac.jp (T.H.); 2Institute of Industrial Science, The University of Tokyo, Tokyo 153-8505, Japan; ta-osawa@iis.u-tokyo.ac.jp (T.O.); yukkam@iis.u-tokyo.ac.jp (Y.K.)

**Keywords:** vibration-induced flow, micro-pillar, numerical analysis, micro-PIV, acoustofluidics

## Abstract

The steady streaming (SS) phenomenon is gaining increased attention in the microfluidics community, because it can generate net mass flow from zero-mean vibration. We developed numerical simulation and experimental measurement tools to analyze this vibration-induced flow, which has been challenging due to its unsteady nature. The validity of these analysis methods is confirmed by comparing the three-dimensional (3D) flow field and the resulting particle trajectories induced around a cylindrical micro-pillar under circular vibration. In the numerical modeling, we directly solved the flow in the Lagrangian frame so that the substrate with a micro-pillar becomes stationary, and the results were converted to a stationary Eulerian frame to compare with the experimental results. The present approach enables us to avoid the introduction of a moving boundary or infinitesimal perturbation approximation. The flow field obtained by the micron-resolution particle image velocimetry (micro-PIV) measurement supported the three-dimensionality observed in the numerical results, which could be important for controlling the mass transport and manipulating particulate objects in microfluidic systems.

## 1. Introduction

The hydrodynamic phenomenon known as steady streaming (SS) is gaining increased attention for controlling flows and associated transport and mixing of chemical species as well as micro objects such as functionalized particles and cells in microfluidic devices [1,2,3,4,5,6,7,8,9,10,11,12]. This term represents the time-averaged non-zero mean flow induced by relative periodic oscillation with zero mean between the substrate and the adjacent bulk fluid. When an infinite planar substrate is oscillating in parallel to its surface, only a transient velocity field with zero mean is generated within a thin layer whose length scale is characterized by the Stokes boundary layer thickness represented as *δ_s_* ~ (2*ν*/*ω*)^1/2^, where *ν* and *ω* are the kinematic viscosity and angular frequency, respectively [13]. However, when an obstacle is present in the flow field, the interaction between the oscillating bulk fluid and the obstacle creates vorticity and causes net-momentum transfer. This is similar to the Reynolds shear stress that arises from the correlation of the fluctuating velocity components in turbulent flows, and it appears as an additional forcing term in the averaged Navier‒Stokes equations. As a result, the steady time-averaged velocity is generated, although the applied periodic forcing does not have a mean component. Because it requires no net displacement or a pressure gradient to drive the flow, the SS is expected to simplify and miniaturize microfluidic systems without introducing external pumps or tubing.

Despite its simplicity, prediction of the flow field induced by SS is nontrivial. Classically the SS flow fields around simple objects, such as a sphere and a cylinder, induced by the oscillatory motion relative to a surrounding fluid were studied [14,15]. The analytical solutions can be obtained in these cases. However, for practical microfluidics applications, a situation could be more complicated; the presence of (often non-straight) channel walls and obstacles with complex shapes does not allow us to derive analytical solutions. To date, several research groups have tackled this problem using numerical analysis. The group of Dr. Schwartz carefully studied the uni-directional oscillating flow in a straight channel with the rectangular cross section, in which an array of cylindrical posts is placed [16,17]. For the numerical analysis, they employed a perturbation approach under two-dimensional flow assumption, in which the periodically oscillating and steady flows were solved separately. Their experimental and numerical results showed good agreement in terms of the streamlines. However, since their main interest was in predicting the location of the center of eddies, into which small particles were trapped, the detailed velocity profile was not fully examined. One of the co-authors of the present work (Dr. Hayakawa) showed that circular vibration, instead of unidirectional vibration, induced the circularly rotating mean flow around a cylindrical micro-pillar. He and his colleagues utilized this phenomenon for manipulation and trapping of cells [7,8,9]. They numerically calculated the flow field using the perturbation approach under the two-dimensional assumption, and obtained peak velocity values in the profile that matched the experimental observations. In addition, the group of Dr. Costanzo established a numerical model to predict the flow in the acoustic-driven micromixing device developed by Dr. Huang in the same university [18,19,20]. Since the frequency of acoustic excitation is higher than SS, they considered the compressibility of the fluid, but the model was based on the perturbation approach under the two-dimensional assumption.

Although numerical results in the abovementioned studies were able to reproduce the experimentally observed SS flow fields, they are based on two assumptions; (1) the amplitude of perturbation (*s*) is small compared to the characteristic length of the system (e.g., the radius of the cylinder *a*), and (2) the flow is two-dimensional. However, the practical operational conditions of SS device may not be limited to the assumed condition of *s*/*a* << 1. Furthermore, in most microfluidic devices, the length-scale perpendicular to the substrate (e.g., the height of channels and structures) is generally much smaller than that in horizontal directions (e.g., the width and length of channels). In such cases, the Stokes layers developing from both top and bottom boundaries are not negligible, so that two-dimensional flow assumption breaks down in most of the flow domain [21]. The three-dimensionality of the SS flow should also arise when the obstacles have 3D shapes. So far, the assessment of the numerical results has mainly been limited to qualitative comparison to the streamlines, which can be readily obtained by the long-time exposure images of the tracer fluorescent beads in experiments. However, the streamlines as well as the magnitude of velocity should strongly depend on the height. To fully predict the SS flows and optimize them, an analytical tool that can directly simulate the 3D field without assumptions is needed.

More recently, several groups reported the 3D numerical simulations of SS flows. Amit et al. calculated the flow around a moving boundary by a commercial solver based on the finite element method [22]. The comparison between the numerical results and the experimental particle image velocimetry (PIV) measurement around a vibrating long cantilever showed quantitative agreement of the velocity field. Rallabandi et al. analyzed the 3D flow field induced by the acoustic actuation of a microbubble, and verified the results through comparison with astigmatism particle tracking velocimetry (APTV) measurements, although the analysis was still based on the small perturbation assumption [23].

Based on the above background, we developed a versatile numerical tool to calculate the 3D SS flow without assuming small perturbation and two-dimensionality of the flow in order to examine the flow induced around a circularly vibrating cylinder placed in a quiescent fluid between two parallel substrates. We employed a Lagrangian approach, in which the coordinate system of the simulation is fixed to the moving (vibrating or oscillating) substrate, instead of an Eulerian approach, in which the boundary is moving relative to the stationary reference frame. The governing equations of an incompressible fluid, i.e., Navier‒Stokes and continuity equations, in which a temporally periodic inertia force due to the circular vibration was included, were directly solved by a pseudo-spectral method [24]. Once the numerical results in the moving (Lagrangian) coordinate were obtained, the flow field was converted to the stationary (Eulerian) coordinate. This approach enabled us to calculate the SS flow field without imposing the moving boundary nor the small perturbation approximation. A fluid‒solid boundary is expressed by the level-set function [25], which is defined as a signed distance function from the surface. This allows to immerse an arbitrary 3D shape in the Eulerian coordinate system and a no-slip condition at a solid surface is achieved by a volume penalization method (VPM) [26]. Such an approach has a great advantage in implementing arbitrary complex obstacles in the fluid domain without generating numerical grids for each geometry. After obtaining the periodically varying instantaneous velocity field, the time-averaged velocity field of Stokes drift was obtained by tracking the virtual fluid particle imposed within the fluid.

The 3D SS velocity field obtained by the numerical simulation was validated by quantitatively comparing with the 3D micro PIV measurement results obtained from the confocal microscopy equipped with a high-speed camera. Our results exhibited the good agreement even within the steeply varying velocity profile within the Stokes layer. The 3D paths of tracer particles induced by the vortex structure adjacent to the edge of the cylinder observed in the experiment were reproduced in the numerical simulation.

## 2. Numerical Procedures

### 2.1. Computational Domain and Governing Equations

In this study, we calculated the flow field around a cylindrical micro-pillar placed between two parallel plates (Figure 1a). Both plates and a pillar fixed to the bottom plate periodically oscillate following a circular path parallel to the substrate (Figure 1b). To simulate this system, we employed a moving coordinate system that moves with the plates instead of the stationary coordinate system with moving boundaries. The flow field obtained in the moving coordinate system was eventually converted to the coordinate system at rest. The advantage of this approach is that the calculation code is simple and numerical accuracy is high, since boundaries (the substrate and the pillar) are stationary with respect to the coordinate system. The effect of the circular vibration was considered by applying an inertia force that rotates in accordance with the acceleration/deceleration of the moving coordinate.

Assuming that the liquid is incompressible and a Newtonian fluid, the liquid flow is governed by the following Navier‒Stokes and the continuity equations:(1)$$\frac{\partial {\underline{u}}_{i}{}^{*}}{\partial t^{*}}+{\underline{u}}_{j}{}^{*}\frac{\partial {\underline{u}}_{i}{}^{*}}{\partial {\underline{x}}_{j}{}^{*}}=-\frac{1}{\rho^{*}}\frac{\partial {\underline{p}}^{*}}{\partial {\underline{x}}_{i}{}^{*}}+\nu^{*}\frac{\partial^{2}{\underline{u}}_{i}{}^{*}}{\partial {\underline{x}}_{j}{}^{*}\partial {\underline{x}}_{j}{}^{*}}$$
(2)$$\frac{\partial {\underline{u}}_{i}{}^{*}}{\partial {\underline{x}}_{i}{}^{*}}=0$$

Here, variables with an asterisk represents a dimensional quantity. The fluid kinetic viscosity and density are denoted by *ν* and *ρ*, respectively. The two orthogonal directions tangential to the substrate are set as *x* and *z*, while the wall-normal direction is *y*. The origin is located at the center of the bottom substrate. Time is defined as *t*. The under-bar indicates a physical quantity in the coordinate system at rest, which is introduced so as to be distinguished from the moving coordinate along a circular orbit defined later. The fluid velocity and static pressure are denoted by ${\underline{u}}_{i}$ and *p*, respectively, where the subscript of *i* represents the three directions, namely, *i* = 1, 2, 3 corresponds to *x*, *y*, *z*, respectively. A micro-pillar was attached to the bottom substrate and its height was set to be equal to the channel half height *δ*^*^. No-slip conditions were applied at the bottom and top walls, i.e., *y*^*^ = 0 and 2*δ*^*^, as well as the surface of the pillar. Periodic boundary conditions were employed in the *x* and *z* directions. This condition corresponds to the case where the geometry shown in Figure 1a repeats in these two directions. The present configuration is chosen because it is relatively simple (the two parallel walls and the periodic placement of cylinders between the walls), while the truncation of the cylinder at the middle of the channel causes complex three-dimensionality of the resulting flow.

In the present study, the solid substrate with micro-pillars oscillates along a given circular tangential orbit as shown in Figure 1b. When the radius and period of the circular oscillation are expressed by *A*^*^ and *T*^*^, the relative displacement of the substrate ***x***^*^*_R_* is given as
(3)$$x_{R}^{*}=\left(\begin{array}{c} x_{R}^{*}\\ y_{R}^{*}\\ z_{R}^{*} \end{array}\right)=\left(\begin{array}{c} A^{*}\cos\left(\frac{2\pi t}{T^{*}}\right)\\ 0\\ A^{*}\sin\left(\frac{2\pi t}{T^{*}}\right) \end{array}\right)$$

Accordingly, the velocity of the solid substrate, $u_{R}^{*}$, can be obtained from the time derivative of ***x***^*^*_R_* as
(4)$$u_{R}^{*}=\left(\begin{array}{c} \frac{dx_{R}^{*}}{dt}\\ \frac{dy_{R}^{*}}{dt}\\ \frac{dz_{R}^{*}}{dt} \end{array}\right)=\left(\begin{array}{c} -\frac{2\pi A^{*}}{T^{*}}\cos\left(\frac{2\pi t}{T^{*}}\right)\\ 0\\ \frac{2\pi A^{*}}{T^{*}}\sin\left(\frac{2\pi t}{T^{*}}\right) \end{array}\right)$$

Solving Equations (1) and (2) under the boundary condition of Equation (4) on the surfaces of the top and bottom substrates and the pillar requires the treatment of moving boundaries. In order to avoid it, a new coordinate system $x^{*}={\left(x^{*},\;y^{*},\;z^{*}\right)}^{T}$, which moves with the same speed as the substrate, is introduced. With the reference length scale of *δ*^*^ and the oscillation period of *T*^*^, the generalized dimensionless forms of Equations (1) and (2) on a rotating frame are expressed as
(5)$$\frac{\partial u_{i}}{\partial t}+u_{j}\frac{\partial u_{i}}{\partial x_{j}}=-\frac{\partial p}{\partial x_{i}}+\frac{1}{Re}\frac{\partial^{2}u_{i}}{\partial x_{j}\partial x_{j}}+f_{i}$$
(6)$$\frac{\partial u_{i}}{\partial x_{i}}=0$$
where the fluid velocity relative to the substrate motion is defined as $u_{i}={\underline{u}}_{i}-u_{R_{i}}$. The effect of the rotational vibration appears as a dimensionless inertia force *f_i_* due to the acceleration/deceleration of the moving coordinate, which is given by
(7)$$f_{i}=\frac{2\pi }{St}\left(\begin{array}{c} \cos\left(2\pi t\right)\\ 0\\ \sin\left(2\pi t\right) \end{array}\right)$$
The detailed derivation of Equations (5)–(7) can be found in the Appendix A.

Here, the Strouhal number *St* is defined as the ratio of the time scales of the flow and the oscillation. Considering that the time scale of flow is given by $\delta^{*}/U_{max}^{*}$, where $U_{max}^{*}=2\pi A^{*}/T^{*}$ is the maximum velocity of the substrate, the Strouhal number is given by
(8)St=δ*Umax*T*=δ*2πA*

The Reynolds number is given by
(9)$$Re=\frac{\delta^{*}{}^{2}}{\nu^{*}T^{*}}$$

The above two dimensionless parameters characterize the flow field considered in the present study. Since we solved the velocity field in the reference frame moved with the substrate, the top and bottom walls as well as the micro-pillar stay at rest, and therefore all the velocity components on these boundaries became null. In order to impose a no-slip condition at a solid surface with arbitrary geometry, we used a volume penalization method introduced in the next subsection.

### 2.2. Volume Penalization Method

One of the main objectives in the present study is to develop a numerical code that is capable of simulating a flow around pillars with arbitrary shapes. A volume penalization method is a kind of immersed boundary techniques, in which a complex structure is embedded in the Cartesian coordinate system. In contrast to using a boundary-fitted coordinate system, the immersed boundary technique has the advantages that grid generation is quite straightforward regardless of the complexity of the geometry and highly accurate discretization schemes developed for the Cartesian grid system can be applicable.

In the present study, the geometry of the pillar was first expressed by a level-set function in the Cartesian coordinate system. The level-set function $\phi_{0}$ is a signed distance function from a surface [25] and it has been widely used to represent complex interface geometry. Here, $\phi_{0}$ was defined positive inside the solid, and negative in the fluid domain. Then, we converted $\phi_{0}$ to the phase-identification function ϕ, which was ϕ=0 inside the fluid, whereas ϕ=1 in the solid. In order to avoid numerical instability, the phase-identification function smoothly changes from zero to one across the interface within a few grid points. Specifically, the level-set function is converted to the phase-identification function by the following formulas:(10)$$\phi =0\;\phi_{0}<-\delta_{int}$$
(11)$$\phi ={\left[1+\exp\left\{\frac{4\left(\phi_{0}/\delta_{int}\right)}{{\left(\phi_{0}/\delta_{int}\right)}^{2}-1}\right\}\right]}^{-1}\;-\delta_{int}<\phi_{0}<\delta_{int}$$
(12)$$\phi =1\;\delta_{int}<\phi_{0}$$
The above function was chosen because it is differentiable within the entire domain, while it is exactly zero and unity in the fluid and solid domain, respectively.

In the volume penalization method, a no-slip condition at the solid surface is realized by introducing an artificial damping force to the Navier‒Stokes equation (Equation (5)) as follows:(13)$$\frac{\partial u_{i}}{\partial t}+u_{j}\frac{\partial u_{i}}{\partial x_{j}}=-\frac{\partial p}{\partial x_{i}}+\frac{1}{Re}\frac{\partial^{2}u_{i}}{\partial x_{j}\partial x_{j}}+f_{i}-\eta \phi u_{i}$$
Here, the final term on the right-hand side is the volume penalization term. Obviously, this term has a non-zero value and acts to suppress all the velocity components inside the solid, while Equation (13) reduces to the original Navier‒Stokes equation (Equation (5)) within the flow domain where ϕ=0.

The advantage of the volume penalization method is that solid objects with different shapes can be easily implemented by changing the spatial distribution of ϕ in the same Cartesian grid system. The drawback is that the grid convergence is relatively slow, since the interface is not explicitly captured and smeared within a few grid points as mentioned above. In the present study, we made grid convergence tests and confirmed that the present results do not change significantly by refining the mesh further (Appendix A).

### 2.3. Numerical Methods and Conditions

We solved Equations (6) and (13) to obtain the velocity field in the moving coordinate by a pseud-spectral method, in which a solution was expanded by Fourier modes in the *x*, *z* direction and Chebyshev polynomials in the *y* direction, respectively. For time advancement, the Crank‒Nicolson was used for diffusion terms, whereas the second-order Adams Bashforth scheme is applied for the convection terms. The inertia and VPM terms appearing in the third and fourth terms are taken into account with the Euler explicit method. The preset code was validated and successfully applied to control and estimation of unsteady turbulent flows in previous studies [24,27].

In this study, we set the diameter and height of micro-pillar to be 200 and 100 µm, respectively. The diameter of the pillar was shown to have only minor effect on the radial profile of the induced flow using the perturbation theory [8], so we studied the flow around the pillar with this representative diameter. The width of the computational domain was 4*δ*^*^ = 800 µm, which is equal to the spacing of pillars in the experiment. This dimension was set to be large enough so that induced flow profiles of neighboring pillars do not interact. The height of the domain was 2*δ*^*^ = 200 µm. The numbers of modes employed in the current simulation were (*N*_1_, *N*_2_, *N*_3_) = (64 × 33 × 64) in *x*, *y*, *z* directions, respectively. 3/2 rule was used for removing aliasing errors, so that the non-linear terms were evaluated in 1.5 times finer physical grid points in each direction. Throughout this work, the vibration amplitude was *A* = 4 µm and frequency was *f* = 1000 Hz. Accordingly, dimensionless numbers were *Re* = 10 and *St* = 12.4/π, respectively. The numerical time step was set to be Δ*t* = 1.0 × 10^−4^, which indicates that *t*^−1^ = 10^4^ time steps are required to compute the velocity field for one oscillation period (*t* = 1). The computation was started from a stationary flow at *t* = 0, and the rotational vibration was applied for *t* = 60 to achieve a fully developed velocity field. After the transient period, the flow field became completely periodic in one oscillation cycle. All statistics shown below were obtained by integrating the velocity data over one oscillation period after the flow field had reached the statistically steady state.

### 2.4. Derivation of Steady Streaming Flow (Time-Averaged Velocity Field)

It is widely known that the SS flow field could be essentially different depending on whether the averaging is made at a fixed stationary location (Eulerian frame) or along a particle moving with the local fluid velocity (Lagrangian frame) due to its oscillatory nature [20,23]. Therefore, we examined and compared the time-averaged SS velocity fields obtained in the Eulerian and Langrangian frames. In the Eulerian approach, the average velocity field was calculated by simply averaging the vector at identical positions in the stationary coordinate system converted from the moving coordinate system. In the Lagrangian approach, trajectories of virtual fluid particles initially located at a uniform spacing of Δ = 8.33 μm in the 3D flow field were calculated using the 4th order Runge‒Kutta method [28]. The instantaneous velocities between the grids were linearly interpolated from the velocity of the surrounding 6 grid points. The instantaneous velocity field data during one circulating period consisted of 50 time frames, and the velocities at the time points between frames were also linearly interpolated using two neighboring frames. After tracking for five periods of the rotational vibration, the velocity field was obtained from the displacement vectors that connect the start and end points at the identical phase.

## 3. Experimental Procedure

### 3.1. Fabrication of Micro-Pillar Array

We fabricate the 5 × 5 array of cylindrical pillars with 200 μm diameter and 100 µm height, with 800 μm center-to-center intervals in accordance with the numerical simulation described in Section 2, using poly-dimethylsiloxane (PDMS) as the material (Figure 2). Four cylindrical pillars with 1.4 mm diameter and 200 μm height were arranged at the four corners of the substrate as spacers to determine the height of the fluid volume. In practice, the master mold was fabricated on a 2-inch silicon wafer by the deep reactive ion etching apparatus (RIE-400iPB, Samco, Japan) using the Bosch process at a rate of 0.4 μm/cycle. PDMS resin (KE-106, Shin-Etsu Chemical, Japan) mixed with its curing agent at 10:1 weight ratio was poured into the master mold. After curing at 50 °C for 120 min, the PDMS substrate was obtained by peeling it off from the mold (Figure 2b).

### 3.2. Experimental Setup and Conditions

Since the PDMS is hydrophobic, air bubbles are often trapped around pillars when the liquid (water) is dropped on its surface. Thus, the substrate was made hydrophilic by the oxygen plasma (SEDE-GE, Meiwafosis Co., Ltd., Japan) treatment for 5 min. Immediately after this treatment, 10 μL of deionized (DI) water containing 0.5 μm yellow-green fluorescent beads (F8813, Thermo Fisher Scientific Inc., MA, USA) as a tracer was dropped to the center of the substrate. Then, the substrate was covered with a cover glass. The thickness of the fluid layer was set to 200 μm by spacers (Figure 3a). Then this assembly was fixed to the XY piezo-drive stage (ML-20XYL, MESS-TEK, Japan) (Figure 3b) using small slips of double-sided tape.

To generate the circular vibration, sinusoidal wave signals with 90° phase offset was applied to the piezo stage using the waveform generator (AG 1022F, OWON, China) via the amplifier (M 2501-1, MESS-TEK, Japan). Applied voltage at 60 V induced *A* = 4 μm displacement over a wide range of frequency below the resonance of this actuator, which was confirmed from the image of the high-speed camera (Mi-2000, Photron, Japan). The vibration frequency was set at *f* = 1000 Hz throughout the present study in accordance with the numerical simulation.

### 3.3. PIV System

We used the confocal micro-PIV technique to measure the flow field around a micro-pillar. Figure 4 shows the schematic diagram of the confocal micro-PIV system. In this system, sequential images of fluorescence tracer particles are obtained by a high-speed camera (Mi-2000, Photron, Japan) via the high NA objective water immersion lens (XLUMPLFLN 20 XW, OLYMPUS, Japan) and a confocal scanner (CSU-X1, YOKOGAWA, Japan), and are stored in the PC. The continuous wave (CW) blue laser (488 nm, Sapphire SF, COHERENT, CA, USA) was used as the illumination light source. Since the frequency of the micro-pillar was 1000 Hz, the shutter speed was set to 1/21,000 s (~48 μs) so that the instantaneous particle images could be resolved without blurring. The frame rate of the image acquisition was set to 2000 fps.

### 3.4. PIV Analysis

Based on the acquired images, we obtained a two-dimensional velocity field using PIV analysis software (Koncerto II, Seika Digital Image, Japan). For obtaining the velocity vector, a recursive cross-correlation method was used, with a 8 × 8 pixel interrogation window and 50% overlap. This window size corresponds to 6.4 μm × 6.4 μm in the physical dimension. In the post processing, standard deviation validation and median filter were used to remove incorrect vectors. Since the micro-pillar oscillates at 1000 Hz, two images were recorded during the one cycle of a pillar rotation when the frame rate was 2000 fps. The SS velocity field was obtained from the displacement of tracer images at every two frames; i.e., images at identical rotational phase. Finally, the average velocity field induced around the pillar was obtained by averaging the flow fields of the 60 rotational cycles. We confirmed that the resultant average profile was almost independent of the interrogation window size (Appendix A).

### 3.5. Horizontal Visualization

A horizontal view of the trajectory of tracer particle was obtained through the objective lens with a long working distance (PAL-10-A, x10, SIGMAKOKI CO. LTD., Japan; WD = 34 mm) located on the side of the pillar substrate and captured by a monochrome CCD camera (BFS-U3-32S4M-C, FLIR Systems, Inc., OR, USA). Polystyrene beads with 10 μm diameter (01-00-104, micromod Partikeltechnologie GmbH., Germany) were used as tracer particles for easy visualization with low magnification lens. The sedimentation velocity of 10 μm bead with a specific density of 1.03 in water is estimated to be around 6.5 μm/s, so that the sedimentation distance within 1.7 s (duration of observation) is not significant.

## 4. Numerical Results

### 4.1. Instantaneous Velocity Field

In the present numerical simulation, a 3D instantaneous flow field around the pillar was obtained after converting the results in the moving coordinate to the stationary one. Figure 5 shows a two-dimensional (2D) vector plot in the plane 0.5*δ* away from the non-slip boundary at the bottom (*y* = 50 μm; the center plane of the pillar in *y* direction). The vectors and color maps in Figure 5 represent the flow direction and the absolute values of the horizontal component of the velocity ($\left|V_{hor}\right|=\sqrt{u_{1}^{2}+u_{3}^{2}}$), respectively. The region near the pillar is enlarged while the calculation area is 800 μm for *x* and *z* directions. Figure 5a–d show the instantaneous fields with the phase shift of every 90° (see Appendix A for all velocity fields within one rotation). Each flow field is rotationally symmetric, showing that the flow field reaches a fully developed state. High-speed regions appeared at the front and back faces of the cylindrical pillar in the traveling direction. These high-speed regions separate near the two sides of the pillar and cause a pair of vortex-like structures. The maximum fluid velocity in this region is ~ 25 mm/s, which is comparable to the velocity of oscillation *U_max_* = 2*πAf* ~ 25.2 mm/s. Although not shown in the figure, the comparable speed was also observed at the vicinity of no-slip boundaries (e.g., the upper and lower walls).

### 4.2. Time-Averaged Flow Field

In a periodically vibrating flow field, the time-averaged velocity becomes null if there is no obstacle in the oscillating direction. However, a non-zero net velocity field appears when the obstacle exists as a result of the Stokes drift. As mentioned previously, we examined the two time-averaging approaches, i.e., Eulerian and Lagrangian averaging. In the latter, we calculated the velocity vectors from the displacement of the virtual tracer particles. An example of the 2D trajectory of a tracer, initially placed at (*x*, *y*, *z*) = (150, 50, 0 μm) (horizontally 50 μm away from the side wall of the pillar) is shown in Figure 6. The tracer particle moves along the distorted orbital path, which can be seen as the superposition of the circular periodic motion and the steady translational movement toward the upper left in the figure. When the pillar is absent, the trajectory draws a perfect circle and returns to the original position after one rotation. Due to the existence of the pillar, however, the position in the same phase is shifted after each rotation. This displacement divided by the time of one rotation period corresponds to the mean velocity observed in experimental particle tracking generated in the SS flow field.

The two-dimensional vector plots of the averaged velocity field obtained by Eulerian and Lagrangian methods at the *y* = 50 μm horizontal plane are depicted in Figure 7. The color map represents the absolute value of the horizontal component |*V_hor_*| of the average velocity. It is clear that a net flow is induced around the pillar in both cases. However, the radial peak position of the velocity is closer to the pillar in the former case (*r* ~ 120 μm in the Eulerian method, but *r* ~ 130 μm in the Lagrangian method). Moreover, the peak value is about two times greater in Eulerian method compared to the Lagrangian method. The result clearly indicates that the averaged flow field depends on the averaging methods. Because the mass transport and the paths of suspended molecules/particles are governed by the Lagrangian trajectories, the Lagrangian averaging is necessary to predict the above-mentioned transport phenomena in micro devices. In the following sections, the Lagrangian velocity obtained from the present simulation will be compared with the averaged translational velocity of fluorescence particles in the experiment in order to validate the present numerical code.

## 5. Comparison with experimental results

### 5.1. Results of PIV Measurement

The image of the movement of fluorescent tracer particles obtained at the *y* = 50 µm horizontal plane is shown in Figure 8a. Sixty instantaneous images at an identical rotational phase, captured by a high-speed camera, are superimposed. Although tracer beads actually moved along the rotating trajectory in accordance with the rotational vibration, the net displacement along the pillar sidewall can be clearly seen by connecting the positions at the same phase. The result of PIV analysis, obtained in this series of images, is depicted in Figure 8b. The condition is the same as that in the numerical result shown in Figure 7. The overall profile and the magnitude of the velocity are similar to those obtained in the Lagrangian averaging (Figure 7b).

### 5.2. Comparison of Radial Velocity Profile

To compare the results quantitatively, we plotted the radial profiles of |*V_hor_*| in Figure 9. The profiles were averaged in the azimuthal direction of the pillar. The magnitude of the peak velocity obtained by Eulerian method is twice as large as the other, and the peak position of the velocity is closer to the pillar, as qualitatively seen in Figure 7. On the other hand, the velocity distributions obtained by the Lagrangian method and PIV measurement showed a similar trend in terms of the peak position (*r* = 130 µm; 30 µm away from the pillar wall) and the decay of the profile. There is a difference between the two profiles close to the pillar; this difference could be partly caused by the difficulty in resolving the high-shear region in both simulation and PIV. The size of the single grid in the simulation corresponds to be 8.3 μm, and the size of the window for image correlation in PIV corresponds to be 6.4 μm. Particle images tended to blur in the region close to the wall due to the 3D flow described later. Nonetheless, a good agreement in the decaying profile after the peak supports the validity of the present simulation.

Overall, the significant difference between two averaging methods is reasonable considering that the flow is unsteady. The virtual fluid particles in the flow do not stay at the constant radial position *r* with respect to the center of the pillar. Instead, their radial position changes as they rotated around the reference position (center of the rotational path) at each phase. Thus, the velocity at the actual particle position is different from that at the reference position. As this slight difference accumulates, the time averaged velocity per one cycle becomes different from the average velocity obtained at the stationary coordinates. The present result confirms that it is necessary to use the Lagrangian tracking method to reproduce the unsteady flow around the vibrating pillar.

### 5.3. Comparison of Normal Velocity Profile

Next, we examined the velocity distribution in the vertical plane including the Stokes layer. To do this we conducted the micro PIV measurement from *y* = 0 to 100 μm at 10 μm interval. The plane at *y* > 100 μm was not obtained experimentally due to the limitation in the working distance of the objective. The color maps of |*V_hor_*| in *r*-*y* plane obtained in the numerical simulation (Lagrangian averaging) and the PIV measurement are shown in Figure 10a,b, respectively. They showed a similar trend in terms of the peak position of *r* = 130 μm at 40 < *y* < 80 μm, which slightly moves closer to the pillar near the top (*y* ~ 100 μm). This is attributed to the strong three-dimensionality of the flow around the tip of the pillar, as will be shown later. The magnitude of the velocity decays rapidly as the position gets closer to the bottom wall.

The horizontal velocity distributions in the vertical (*y*-) direction at *r* = 130 µm are plotted shown in Figure 11. Both profiles agreed well, with velocity values close to 0 μm/s on the substrate surface (*y* = 0 μm), which increased to ~1.8 mm/s towards the tip of the pillar. The peak value is obtained in the vicinity of *y* = 70 to 80 μm.

In the Stokes second problem of a unidirectional oscillation of the flat solid wall in contact with the fluid, the fluid in the vicinity of the wall almost follows the wall movement, but this influence exponentially decays as it moves away from the wall with a characteristic length *δ_s_* = (2*ν*/*ω*)^1/2^ [13]. From its analytical solution, the effect of the wall motion decreases down to ~5% (1/*e*^3^) at *y* = 3*δ_s_*. In the present condition, 3*δ_s_* corresponds to 53.4 µm by substituting *ν* = 1.0 × 10^−6^ m^2^/s (25 °C) and *ω* = 2π*f* = 6.28 × 10^3^ rad/s. The height is indicated by dotted lines in Figure 10 and Figure 11. Since the net flow discussed in the present system is induced by the relative velocity between the fluid and the substrate, the maximum average flow field occurs outside the Stokes layer. In contrast, the fluid within the Stokes layer thickness 3*δ_s_* is dragged by the motion of the wall due to the viscous effect so that the relative velocity diminishes on the bottom wall.

The present results indicate that the viscous effect adjacent to the solid wall has a strong impact on the net travel distance of a passive tracer, and the thickness of this fluid layer is determined by the Stokes layer thickness. It should be also noted that, for each height *y*, the peak of the averaged velocity occurs about 30 μm away from the side wall of the pillar as shown in Figure 10. This horizontal gap is also similar to the Stokes layer thickness. When SS is applied to micro devices, the Stokes layer thickness is not negligibly small in general, and this causes the strong dependency of the induced averaged flow on the vertical location, i.e., the distance from the top and bottom substrate. Therefore, in order to predict and design the entire velocity field inside a micro device, 3D analysis is requisite.

### 5.4. Three-Dimensionality of the Flow

Lastly, we investigated the three-dimensionality of the averaged velocity field around the pillar. Figure 12 shows a vector and a contour plot of the averaged velocity in the vertical plane ($\left|V_{ver}\right|=\sqrt{u_{1}^{2}+u_{2}^{2}}$) obtained from the numerical simulation. In the vicinity of the upper and lower wall surfaces, it is clear that there is almost no upward/downward motion, so that the flow is practically two-dimensional. However, at the intermediate region between the top and bottom plates, the velocity toward the pillar with the magnitude as large as ~200 μm/s appears around the corner of the pillar (*r* = 130 to 150 μm and *y* = 50 to 100 μm). This flow is diverted upward at the pillar edge. As a result, a vortical motion is generated close to the corner of the pillar. Although the magnitude is about 10 times smaller than |*V_hor_*|, the presence of the vertical velocity component has a strong impact on the averaged velocity field as discussed in Figure 10.

To confirm the three-dimensionality of the averaged velocity flow field in the experimental flow field, the motion of the particle was observed from the side of the pillar using an objective lens with a long working distance placed horizontally. Because the confocal illumination is not possible away from the wall, we introduced 10 μm polystyrene beads as tracers into the fluid at a low concentration so that individual beads were visible from a long distance. Two representative paths visualized by superimposing ~50 successive frames (1.7 s) are shown on the left column in Figure 13a,b (see Appendix A for corresponding movies). In both cases, tracer particles exhibited three-dimensional motions including upward and downward movements, instead of a simple orbital movement in the same horizontal plane. In Figure 13a, a bead initially circulating around the middle of the pillar was suddenly raised at the region close the apex, and then hovered above the pillar. The similar path could be reproduced by tracking an ideal tracer in the averaged velocity field obtained by the Lagrangian averaging (Figure 7b and Figure 12) in the simulation (right figures). This sudden ascension should be caused by the upward flow near the apex discussed in Figure 12. Another bead shown in Figure 13b, which was circulating a bit far outside of the pillar slightly, shifted downward as it got closer to the pillar, and then raised. This path was also reproduced in the numerical simulation (right figures), which was convected by the downward net velocity at *r* = 120 ~ 170 μm and *y* = 100 ~ 130 μm, and raised by the upward net velocity at *r* = 110 ~ 150 μm and *y* = 40 ~ 100 μm shown in Figure 12. These trajectories show the validity of numerical prediction for three-dimensionality of flow induced around the pillar at the present vibration condition.

## 6. Conclusions

In this work, we developed a simulation tool to predict the SS flow induced by vibration without assuming 2D flow and small vibration amplitude. The developed numerical code was based on the volume penalization method in the Cartesian coordinate system, so that arbitrary three-dimensional structures can be readily embedded without requiring grid generation for each geometry. As pointed out in the past SS studies, we confirmed that the average velocity at fixed points (Eulerian mean) and that of fluid particles advected by the local flow velocity (Lagrangian mean) are essentially different. In practical microdevice applications, the Lagrangian-averaged flow governs mass transport and mixing. In this work, we compared the Lagrangian mean field obtained by numerical calculation with micro PIV experimental results. The quantitative agreement between them validates the present simulation method.

Through these numerical and experimental analyses, we observed two 3D characteristics of the SS flow even at the flow Reynolds number of 10. First, inside the Stokes layer, which develops from the solid surface toward the fluid region, the averaged velocity due to the Stokes drift diminishes because the fluid is dragged by the solid motion due to the viscous effect, and thereby the relative velocity between the fluid and the solid reduces. Since the thickness of the Stokes layer cannot be neglected with respect to the length scale of microdevices in general, it is necessary to accurately predict the influence of the Stokes layer in order to understand the flow inside the device. Secondly, a large strong vortex motion accompanying upward and downward motions was confirmed in the vicinity of the apex of the micro-pillar. Such vertical fluid motions could have significant influences on mass transport, and may result in a complicated flow field combined with the three-dimensionality caused by the Stokes layer mentioned above. Since the present numerical results reproduce the 3D nature of the flow fields observed in experiment, our numerical code could be a powerful tool to optimize the structure of a micro-pillar and the vibration mode in order to develop innovative microfluidic devices in future work.

## Figures and Tables

**Figure 1 micromachines-09-00668-f001:**
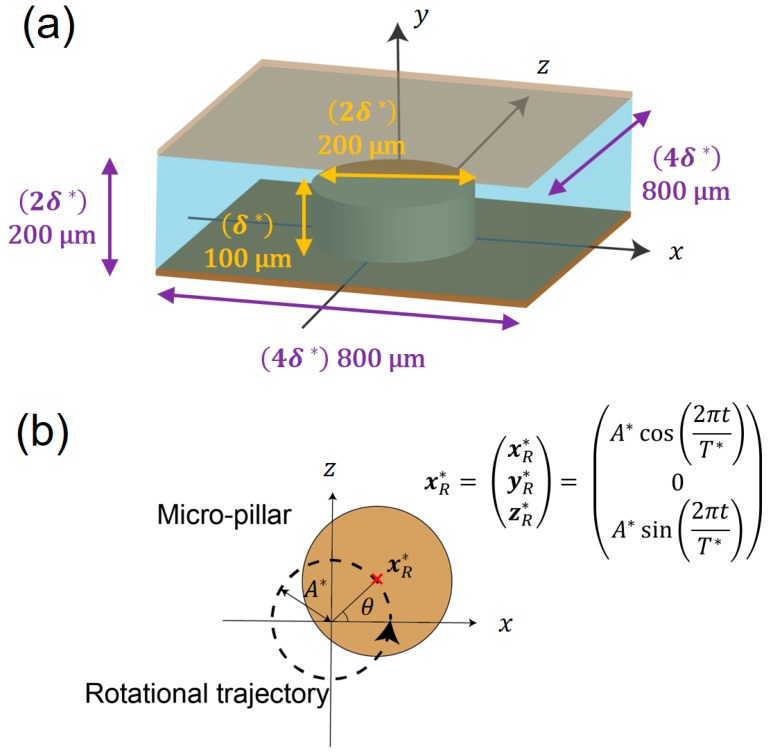
(**a**) Schematic of coordinate system and computational domain. (**b**) Circular vibration of the micro-pillar.

**Figure 2 micromachines-09-00668-f002:**
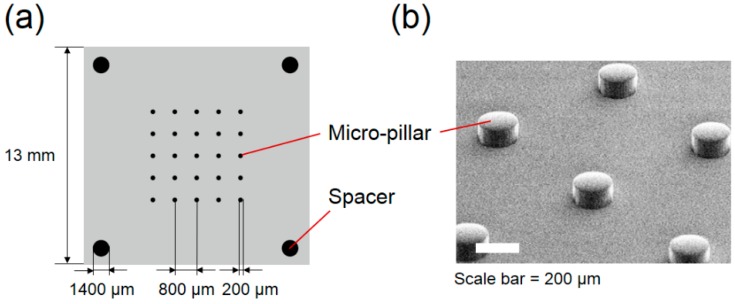
(**a**) Arrangement of the micro-pillars on a substrate. (**b**) SEM image of the micro-pillars.

**Figure 3 micromachines-09-00668-f003:**
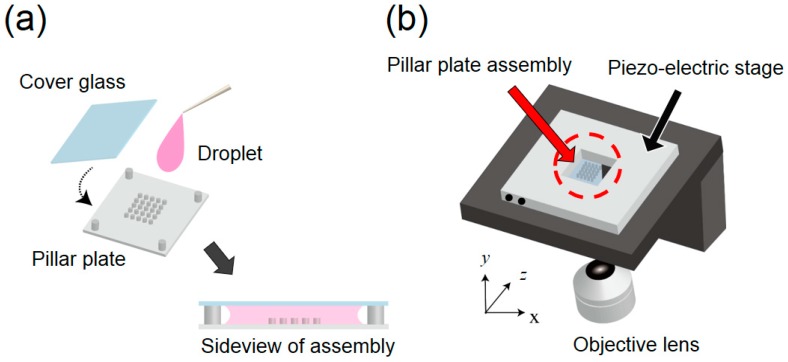
(**a**) Assembly of the micro-pillar plate. (**b**) Experimental setup for application of circular vibration.

**Figure 4 micromachines-09-00668-f004:**
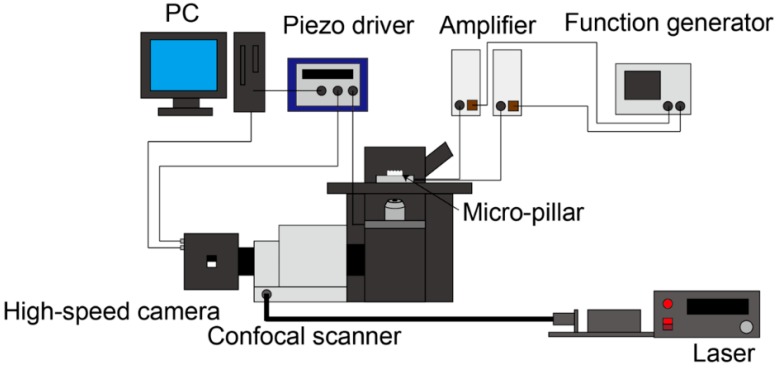
Schematic diagram of the confocal micro-PIV system.

**Figure 5 micromachines-09-00668-f005:**
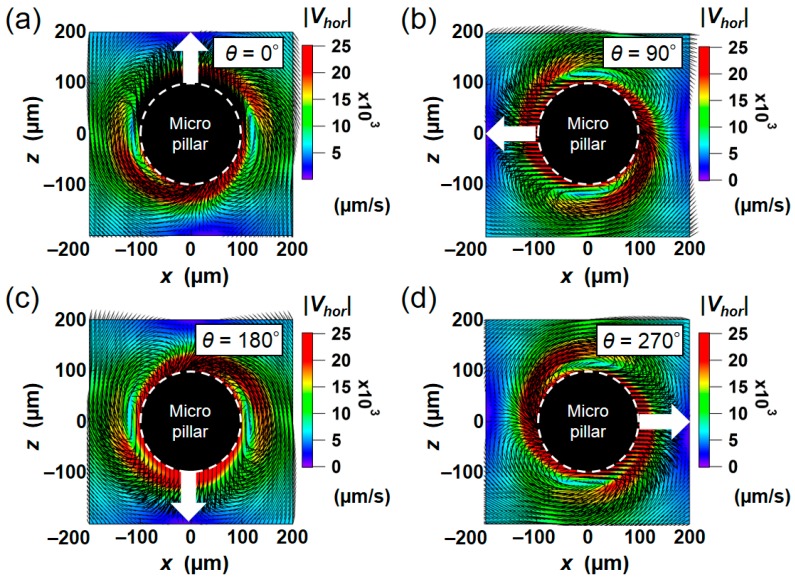
2D vector plot of the instantaneous velocity field at phases of (**a**) 0°, (**b**) 90°, (**c**) 180°, (**d**) 270° within the one rotational cycle. White arrows indicate the instantaneous moving directions of the pillar.

**Figure 6 micromachines-09-00668-f006:**
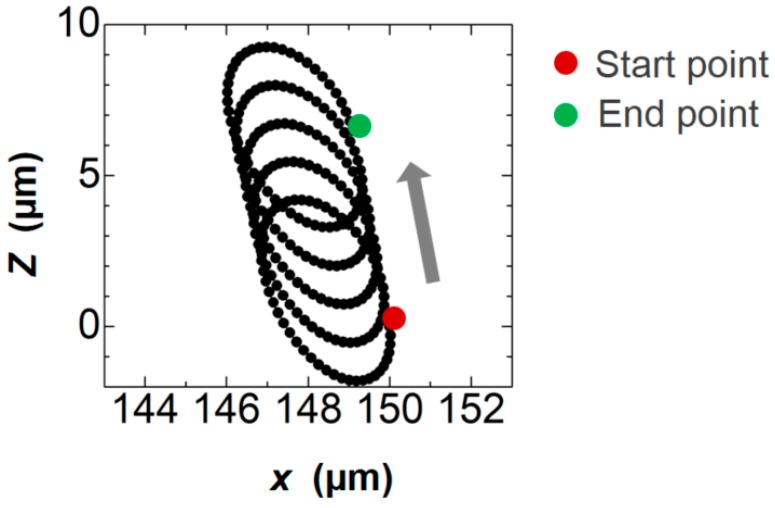
A 2D trajectory of a particle during five rotations.

**Figure 7 micromachines-09-00668-f007:**
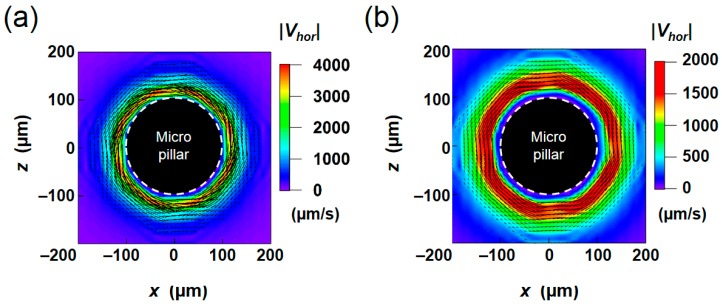
2D vector plots of the averaged velocity flow field calculated by the (**a**) Eulerian and (**b**) Lagrangian approaches.

**Figure 8 micromachines-09-00668-f008:**
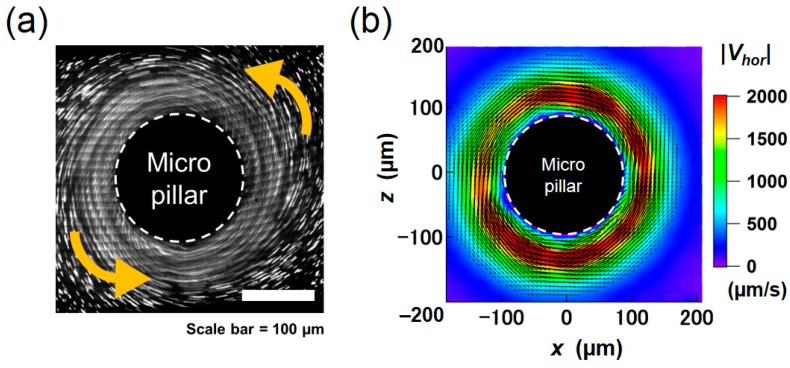
(**a**) Raw image of fluorescent tracer beads overlaid for sixty cycles. (**b**) 2D vector plot of the averaged velocity flow field obtained from PIV measurement.

**Figure 9 micromachines-09-00668-f009:**
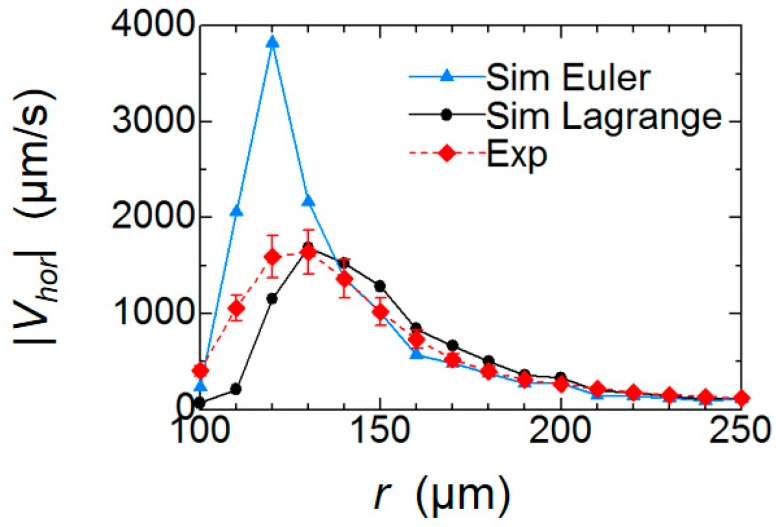
Distribution of the mean horizontal velocity magnitude of particles in radial direction. (▲) Eulerian averaging and (●) Lanrangian averaging of the simulation result. (◆) PIV measurement. PIV was repeated three times with different setup, and the average and standard deviation are shown.

**Figure 10 micromachines-09-00668-f010:**
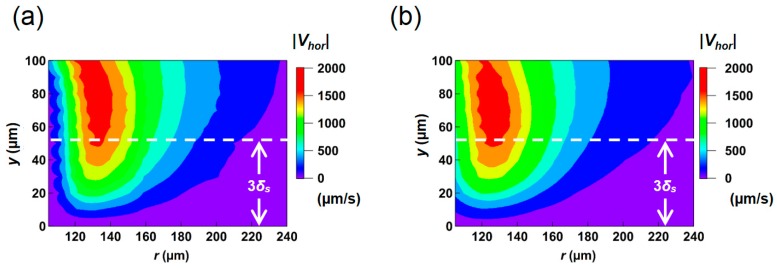
Contour plots of mean horizontal velocity magnitude |*V_hor_*| in *r*-*y* plane of (**a**) numerical result (Lagrangian averaging) and (**b**) PIV measurement. The dashed line indicates the Stokes layer thickness calculated based on the present experimental condition.

**Figure 11 micromachines-09-00668-f011:**
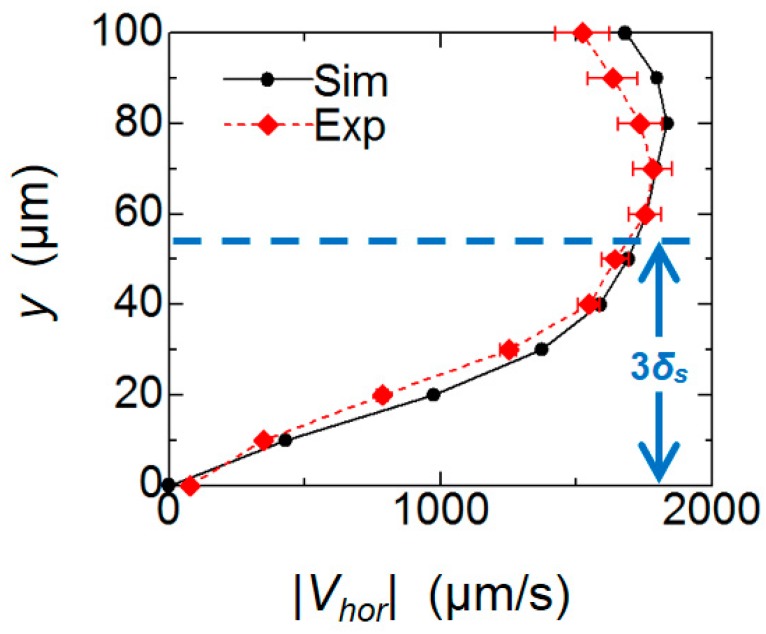
Comparison of vertical distributions of mean horizontal velocity magnitude |*V_hor_*|.

**Figure 12 micromachines-09-00668-f012:**
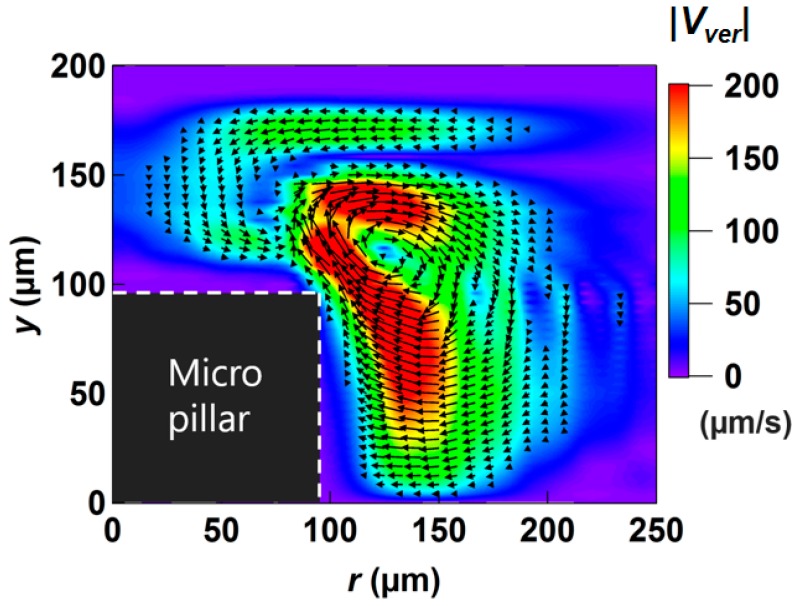
Vector plot of the averaged velocity flow (Lagrangian method) in the vertical plane.

**Figure 13 micromachines-09-00668-f013:**
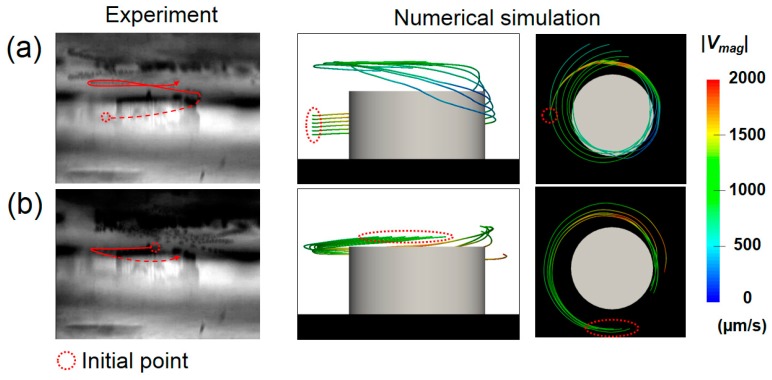
Comparison of particle trajectories obtained in the experiment and the numerical simulation. (**a**) Ascending motion near the apex of the pillar. (**b**) Descending and ascending motion.

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
