# Peer review of "Numerical and Experimental Analyses of Three-Dimensional Unsteady Flow around a Micro-Pillar Subjected to Rotational Vibration"

_micromachines, 2018, doi:10.3390/mi9120668_

Round 1

Reviewer 1 Report

This manuscript describes both numerical and experimental analyses of 3D unsteady flow around a micropillar. In general, the manuscript is well-written and the results are convincing. However, the authors may consider making the minor modification before the manuscript is published.

In section 3, the authors may want to describe in detail the DRIE process. This will allow readers to replicate the experimental setup. 

     2. Add in discussion on the following:

         - what will happen when the size of the pillar is changed?

         - what will happen if the spacing between the pillars are irregular?

         - How does the dimensions of the pillars and spacing affect the flow?

     3. The surface modification of PDMS using plasma does render the surface hydrophilic.                      However, this plasma treatment is transient and will cause the PDMS to revert to hydrophobic after some time. Hence, measuring the contact angle before and after the experiments will help to confirm if the change in wetting properties contributed to the flow. In addition, the authors may want to comment on this also. 

Author Response

Reviewer #1

1. In section 3, the authors may want to describe in detail the DRIE process. This will allow readers to replicate the experimental setup. 

> Thank you for pointing out. We added the name and make of the DRIE apparatus, name of the process, and etching rate (line 290).

2. Add in discussion on the following:

- what will happen when the size of the pillar is changed?
> The diameter of the pillar has only a minor effect on the radial mean velocity profile, which was shown in Fig. 2(c) to (e) in the reference [8] (Hayakawa et al., Microsyst. Nanoeng., 2015).

- what will happen if the spacing between the pillars are irregular?          

- How does the dimensions of the pillars and spacing affect the flow?

> In the present experiment, we set the spacing between pillars large enough so that interaction of flow fields is negligible. When the spacing is smaller than the scale of velocity profile, profile deforms reflecting the arrangement of pillars.

We added explanations for these points in line 241 to 245 in the revised manuscript.

3. The surface modification of PDMS using plasma does render the surface hydrophilic. However, this plasma treatment is transient and will cause the PDMS to revert to hydrophobic after some time. Hence, measuring the contact angle before and after the experiments will help to confirm if the change in wetting properties contributed to the flow. In addition, the authors may want to comment on this also. 

> We added a water droplet immediately after the oxygen plasma treatment (we modified the line 301). Once the PDMS surface was wet without trapped bubbles, wettability of the surface does not affect the flow (it merely works as a nonslip boundary). The experiment was completed within 20 to 30 min.

Reviewer 2 Report

Some generalities

The authors provide numerous equations, especially, between pages 3 and 6.  While I leave it up to the authors to condense this material (with inclusion of appropriate references and without loss of clarity), it appears to me that what is described in these pages is standard textbook material.  Also, Runge-Kutta methods have been around for several decades (and software-routines for them have been published).

The supporting material includes interesting movies, yet, I did not see a mention on the text of this manuscript leading interested readers to the supporting documentation (and these videos).  This I consider a serious omission, and it must be fixed.  Also, on page 16, line 540, there is a pointer to supplementary material, but the link is missing.  This should also be fixed.

Some specifics: While Fig. 1 is first referenced on line 116 (page 5), the actual figure does not appear until the bottom of page 6.  In general, figures should appear very close to the first instance they are mentioned.  This should be fixed.

Reference list: for a 16-page long manuscript, I was expecting a more substantial reference list that included more recent references.  More recent references should be added (otherwise interested readers may get the impression that this field is obsolete).

Research keywords: they should be modified to more accurately reflect the content included in this manuscript, the current keywords do not accurately reflect (and are missleading) based on what was included in this manuscript.

Author Response

Reviewer #2

Some generalities

The authors provide numerous equations, especially, between pages 3 and 6.  While I leave it up to the authors to condense this material (with inclusion of appropriate references and without loss of clarity), it appears to me that what is described in these pages is standard textbook material.  Also, Runge-Kutta methods have been around for several decades (and software-routines for them have been published).

> Thank you for your comment. We moved the derivation of the governing equations in the moving coordinate to the supplementary text, to keep the main text concise.

The supporting material includes interesting movies, yet, I did not see a mention on the text of this manuscript leading interested readers to the supporting documentation (and these videos).  This I consider a serious omission, and it must be fixed. 

> We believe that all supporting materials including videos are properly referred in the original main text.

Also, on page 16, line 540, there is a pointer to supplementary material, but the link is missing.  This should also be fixed.

> We believe that the direct link to the supporting information files will be supplied by the publisher editing team after acceptance. We added the detailed information for the Supplementary Materials section at the end of the revised manuscript.

Some specifics: While Fig. 1 is first referenced on line 116 (page 5), the actual figure does not appear until the bottom of page 6.  In general, figures should appear very close to the first instance they are mentioned.  This should be fixed.

> Thank you for pointing out. We moved Fig. 1 to the beginning of Section 2.1.

Reference list: for a 16-page long manuscript, I was expecting a more substantial reference list that included more recent references.  More recent references should be added (otherwise interested readers may get the impression that this field is obsolete).

> Thank you. We added 8 references, in which acoustic fluid manipulations were used for.

Research keywords: they should be modified to more accurately reflect the content included in this manuscript, the current keywords do not accurately reflect (and are missleading) based on what was included in this manuscript.

> We reviewed the present keywords, and we did not understand why these are not adequate. We appreciate if you could tell us the reason. Any suggestion for new keywords is welcome.

Round 2

Reviewer 1 Report

I am satisfied with the response and would like to recommend the acceptance of the work.